# Study on Comparisons of Bio-Hydrogen Yield Potential and Energy Conversion Efficiency between Stem and Leaf of Sweet Potato by Photo-Fermentation

**Haorui Zhang [1], Tingzhou Lei [2], Shijie Lu [1], Shengnan Zhu [1], Yameng Li [1], Quanguo Zhang [1] and Zhiping Zhang [1,*]**

[1] Henan International Joint Laboratory of Biomass Energy and Nanomaterials, Henan Agricultural University, Zhengzhou 450002, China; zzhr1996@163.com (H.Z.); lushijie1001@163.com (S.L.); zhushengnan0105@163.com (S.Z.); liyameng2017@163.com (Y.L.); zquanguo@163.com (Q.Z.)

[2] Institute of Urban and Rural, Changzhou University, Changzhou 213164, China; 13837166683@163.com

[*] Correspondence: zhangzhipinghau@henau.edu.cn; Tel.: +86-136-7365-3871; Fax: +86-371-6355-8267

**Abstract:** The source of raw materials for hydrogen production can be expanded by using vine waste as a substrate. Likewise, the effectiveness of vine waste can also be improved. However, plant parts such as stems and leaves often differ in physicochemical properties, which significantly affects the effectiveness of biochemical transformation. In this research, sweet potato was used as substrate in photo-fermentative hydrogen production (PFHP) to evaluate differences in bio-hydrogen production yield potential and energy conversion efficiency for its stem and leaf. Physicochemical properties were determined using the following techniques: elementary analysis, SEM, and X-ray diffraction. The Gompertz model was adopted to analyze the kinetic parameters, and energy conversion efficiency was calculated. The results showed that stem samples with loose structures produced more hydrogen, with a total cellulose and hemicellulose content of 44.6%, but crystallinity was only 29.67%. Cumulative bio-hydrogen yield of stem was 66.03 mL/g TS, which was 3.59 times higher than that of leaf. An increase of 258.93% in energy conversion efficiency was obtained when stem was used for PFHP. In conclusion, stem samples were more suitable for PFHP than leaf samples.

**Keywords:** photo-fermentative biohydrogen production; cumulative bio-hydrogen yield; physicochemical property; energy conversion efficiency; sweet potato

## 1. Introduction

The burning of fossil fuels releases greenhouse gases, leading to the greenhouse effect, which has an adverse impact on the environment [1]. There is a growing consensus that energy systems need to achieve net-zero emissions to prevent the damaging effects of climate change [2]. Therefore, the search for a clean alternative to fossil fuels is a key requirement for dealing with the greenhouse effect caused by the growth of industrialization [3]. Hydrogen is suitable as a clean energy alternative due to its high calorific value and production of water as a byproduct of combustion, which thus produces no greenhouse gas emissions [4].

At present, hydrogen is mainly produced by cracking, gasification, and steam reforming of natural gas; water electrolysis; renewable liquid reforming; dark fermentation; photo fermentation; and other methods [5]. Compared to traditional hydrogen production methods, the hydrogen production by biomass fermentation shows greater potential due to the pollution-free combustion of its products. The main processes for hydrogen production by biomass fermentation are dark fermentation and photo fermentation [6]. Although dark fermentation is an easy process to operate, the substrate conversion rate is low. However, photo-fermentative hydrogen production (PFHP) results in high hydrogen yield with minimal nutritional requirements. In addition, photo-catalytic hydrogen production uses solar energy to split water for hydrogen production in order to convert solar energy into chemical energy storage. Thus, compared with PFHP, though both use solar energy, photo-catalytic

hydrogen production reaction equipment is complex, the addition of a catalyst is necessary, and the reaction conditions are harsh [7]. Hence, PFHP is a more promising method for producing $H_2$ [8]. Photosynthetic bacteria produce hydrogen by using organic matter as carbon source and hydrogen donor under light condition. The reaction can be carried out under mild conditions (room temperature and pressure), and various agricultural waste, food waste, etc., can be utilized as raw materials [9].

China has plentiful agricultural waste resources that can be degraded for PFHP. Research of biohydrogen production from corn stover, corncob, sorghum, and energy grass has been conducted [10,11], while there is less literature about PFHP from sweet potato biomass. Sweet potato has a long history and a wide planting range in China. In 2020, China produced 52 million tons of sweet potatoes. It grows fast and has high biomass output. High contents of cellulose and hemicellulose makes it a potential hydrogen-producing substrate. China's sweet potato planting area ranks first in the world. However, most of the stems and leaves left after harvest are discarded and mostly abandoned. Hence, using this lignocellulosic biomass in hydrogen production technology can help with the recycling of biomass waste, which is not only environmentally friendly but also produces hydrogen energy and is a beneficial exploration to expand the source of bio-hydrogen energy raw materials.

Moreover, the physical structure and chemical composition of different parts of the same plant are different [12]. Especially in vines, there are great differences between stem and leaf [13]. Sweet potato also has a long stem and large leaf. Hence, attention should be paid to the difference between sweet potato stem and leaf. Thus, a new substrate species may be provided for hydrogen production for PFHP.

In this work, sweet potato stem and leaf were used as raw materials. The PFHP process was monitored. The kinetic parameters were analyzed by the Gompertz model. Samples were analyzed for differences in physicochemical properties using the following techniques: elementary analysis, SEM, and X-ray diffraction. Moreover, the energy conversion efficiency was also calculated. The comparison of hydrogen yield capacity and energy conversion efficiency between stem and leaf was conducted to observe the differences.

## 2. Materials and Methods

### 2.1. Materials

#### 2.1.1. Raw Materials

The raw materials used in the experiment were sweet potato stems and leaves, which were taken from the western suburb of Kaifeng. Kaifeng, Henan Province, China. The raw materials were dried in an oven to a constant weight at 60 °C and then crushed to powder. After passing 60 mesh sieves, powders of sweet potato stem and leaf were stored in transparent self-sealing bag at normal temperature for later use.

#### 2.1.2. Strains and Medium

Photosynthetic bacteria (PSB) HAU-M1 was obtained from the Key Laboratory of New Materials and Facilities for Rural Renewable Energy, Ministry of Agriculture and Rural Affairs, Zhengzhou, China [14]. The growth medium of HAU-M1 (per 1 L) included NaCl 2 g, $K_2HPO_4$ 0.2 g, $NaHCO_3$ 2 g, $MgSO_4$ $7H_2O$ 0.2 g, $CH_3COONa$ 3 g, $NH_4Cl$ 1 g, and yeast extract 1 g. Each 1 L of the hydrogen producing medium included NaCl 2 g, $MgCl_2$ 0.2 g, $NH_4Cl$ 0.4 g, yeast extract 0.1 g, sodium glutamate 3.5 g, and $K_2HPO_4$ 0.5 g [15]. PSB was cultured under 3000 Lux and 30 °C conditions. The experimental light source came from incandescent lamps.

### 2.2. Experimental Procedures

The enzymatic hydrolysis experiments were conducted in 250 mL conical flasks. Sweet potato stem powder and leaf powder were employed as substrates, while cellulose was employed for enzymatic hydrolysis. In each flask, substrate weight was 3 g, volume citric acid–sodium citrate buffer was 100 mL, and enzyme load was 0.2 mL/g. All flasks were

settled in a thermostatic oscillator for 48 h enzymatic hydrolysis, with temperature of 50 °C and oscillation speed of 150 rpm.

After enzymatic hydrolysis, the hydrolysates were titrated to neutral by adding HCl and KOH solution. Then, high activity HAU-M1, which was at its logarithmic growth phase, was added into each flask with the volume fraction of 30%. Hydrogen producing media were appended according to the proportion mentioned above. A rubber plug was utilized for sealing, and an airbag was employed for biogas collection. The sealed flasks were placed in a constant temperature and humidity incubator. The temperature was set at 30 °C, and the light intensity was adjusted to 3000 Lux. Samples were monitored every 12 h. Three parallel experiments were set for each group to ensure the reliability of data.

### 2.3. Analytical Methods

A pH meter (PHS-3C, Shanghai, China) was used to measure the solution pH. A digital illuminance meter (TES-1330, Shanghai, China) was used to measure the light intensity. The di-nitrosalicylic (DNS) colorimetric method was used to determine the reducing sugar concentration, and the reducing sugar concentration was calculated by optical density at 540 nm. Composition analysis of biogas was detected by using Agilent 6820 GC-14B gas chromatography (USA), including hydrogen concentration. Column packing was equipped with a 5 A molecular sieve. Nitrogen was used as carrier gas, with a flow rate of 45 mL/min. The standard gas used was high purity hydrogen (99.999%). Temperatures of the inlet were 100 °C, column oven was 80 °C, and TCD was 150 °C. A testing volume of 500 μL was used with a retention time of 2 min. An X-ray diffractometer (PANalytical X'Pert PRO, Netherlands) was employed to analyze the X-ray diffraction of samples with Cu Kα radiation at 40 kV and 40 mA, according to the Segal method, calculating the relative crystallinity (*CrI*) of samples using Equation (1) [16].

$$C_r I(\%) \ = \ \frac{I_{002} \ - \ I_{am}}{I_{002}} \tag{1}$$

where $C_r I$ is the relative crystallinity; $I_{002}$ is the diffraction peak of the (002) crystal plane, which is the maximum diffraction peak of X-ray diffraction near $2\theta \approx 22°$; and $I_{am}$ is the non-specular diffraction peak, which is the troughs of X-ray diffraction near $2\theta = 18°$.

A scanning electron microscope (JSM-7500) was used to study the microstructures of the samples. All samples were dried at 60 °C to a constant weight, and samples were affixed using conductive tape. In order to enhance the electric conduction of samples, the samples were sprayed with platinum metal.

The sweet potato stem and leaf after drying used to crush to powder were sifted to below 0.09 mm. The compositions (cellulose, hemicellulose, and lignin) and main elementary composition of substrates were determined by the standard biomass analysis method proposed by the National Renewable Energy Laboratory (NREL) [16].

### 2.4. Kinetic Analysis

In order to investigate the kinetic parameters of PFHP from sweet potato stem and leaf, we used the modified Gompertz model (Equation (2)) [17].

$$P(t) \ = \ P_m exp\left\{-exp\left[\frac{R_m e}{P_m}(\lambda \ - \ t) \ + \ 1\right]\right\} \tag{2}$$

where *P(t)* is cumulative hydrogen yield, mL/g TS; $P_m$ is the maximum hydrogen yield potential, mL/g TS; $R_m$ is the maximum hydrogen production rate, mL/(g TS·h); $\lambda$ is lag time, h; *t* is fermentation time, h; and e is 2.72.

### 2.5. Energy Conversion Efficiency Evaluation

The energy conversion efficiency of the substrate refers to the ratio of the energy converted into hydrogen by the substrate in the hydrogen production experiment to the

total energy of the substrate, where the substrate energy refers to the calorific value of the substrate. The energy conversion efficiency of hydrogen production was calculated using Equation (3) [17].

$$E = \frac{V_{H_2} \times Q_{H_2}}{Q_{net} \times m} \times 100\% \tag{3}$$

where $E$ is energy conversion efficiency, %; $V_{H2}$ is the volume of $H_2$, mL; $Q_{H2}$ is the heat value of hydrogen gas, 12.86 J/mL; $Q_{net}$ is the heat value of the sweet potato stem and leaf, 14,075.70 J/g; and $m$ is the weight of the sweet potato biomass powder, g.

### 2.6. One-Way Analysis of Variance Evaluation

Statistical analyses were carried out using Excel software (Microsoft Inc., Redmond, WA, USA). Correlations between different substrates (stem and leaf) and PFHP were analyzed using the one-way analysis of variance test to check significant differences. Particularly, $p$-value and F-value were evaluated to ascertain significance.

## 3. Results and Discussion

### 3.1. Comparison of the Physicochemical Properties of Sweet Potato Stem and Leaf

Lignocellulosic biomass is the raw material for photo-fermentative biohydrogen production (PFHP). Diverse lignocellulose biomass shows different PFHP performance due to differences in composition and structure. Hence, the elementary content, degree of crystallinity, and microscopic structure and surface morphology of stem and leaf were determined. The SEM images and XRD spectrogram are shown in Figure 1, and the elementary contents of stem and leaf are listed in Table 1.

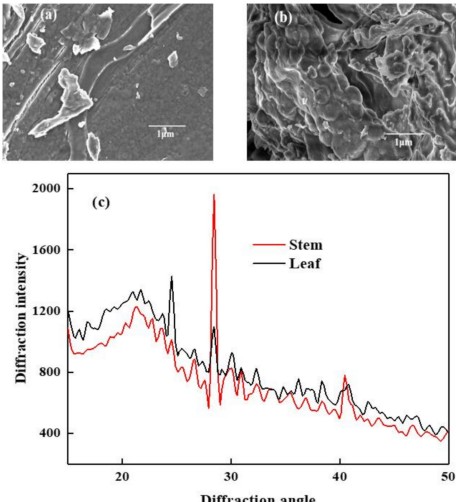

**Figure 1.** SEM images of sweet potato stem (**a**), sweet potato leaf (**b**), and the XRD spectrogram of stem and leaf (**c**).

**Table 1.** Element analysis of sweet potato stem and leaf and the compositions.

| Plant Part | C | H | N | S | Cellulose | Hemicellulose | Lignin | $C_r I$ |
|---|---|---|---|---|---|---|---|---|
| Stem | 3.39% | 38.01% | 4.95% | 0 | 23.4% | 21.2% | 11.4% | 29.67% |
| Leaf | 4.01% | 41.71% | 5.25% | 0 | 22.2% | 16.9% | 13.4% | 21.95% |

As shown in Figure 1, different magnifications of stem and leaf were displayed using scanning electron microscopy. By comparing the SEM results of stem and leaf, the surface of stem was found to be loose and porous, mostly in the shape of fragments and particles, a small part of it was rod-shaped, and the arrangement was disorderly. This structure indicted the barrier breaking of lignin. The degree of hemicellulose covering the fiber

bundles was reduced, which could effectively increase the accessibility of cellulase and facilitate cellulase to open the polysaccharide structure of cellulose to form glucose and xylose [18]. Unlike stem, the surface of leaf was mostly large, with a complete structure that was dense and thick. This structure hindered the accessibility of cellulase and inhibited cellulase from converting polysaccharides into reducing sugars, thereby inhibiting the metabolism and hydrogen production of photosynthetic bacteria.

Analysis of the X-ray diffraction pattern is based on the intensity and position of the strongest point of diffraction to calculate the crystallinity in the molecular chain of cellulose alert. The (002) crystal plane is a commonly used crystal plane for X-ray determination of crystal structures. The diffraction intensity peak of the (002) crystal plane appeared at a maximum peak near $2\theta = 22°$. There was a very small peak near $2\theta = 18°$. Based on this, the relative crystallinity of cellulose could be calculated.

As can be seen in Figure 1c, the diffraction intensity of leaf was greater than that of stem, and the crystal plane intensity of leaf was about 19% higher than that of stem, indicating that the crystallinity of leaf was higher than that of stem. The $C_rI$ values of stem and leaf were calculated to be 29.67% and 21.95%, respectively. Zhang et al. measured the $C_rI$ of raw corn straw in an experiment, which were 32.95% to 36.9% [16], which were higher than the values of the substrate in this paper. Although the $C_rI$ of stem was higher than that of leaf, the diffraction intensity of the crystal plane of leaf was higher than that of stem. As a result, the stem is enzymatically more thorough than leaf, providing more sources for photosynthetic bacteria for hydrogen production [19]. Moreover, the diffraction peak of Si appeared near $2\theta = 27°$; this may be because a small amount of sediment was still left in the process of raw materials pretreatment [16].

As shown in Table 1, carbon and nitrogen sources are the most basic and essential elementary nutrients in microbial metabolism. They are closely related to the metabolic process of microorganisms. The nitrogen source is one of the main factors affecting the hydrogen-producing activity of photosynthetic bacteria. This is because nitrogenase is the main hydrogen-producing enzyme in photosynthetic bacteria. The excited light-harvesting occurs by absorbing light energy and high energy electron reaction center. A part of the electron chain is transferred through photosynthetic phosphorylation and is used to synthesize adenosine triphosphate (ATP) under the action of ATPase; the other part of high-energy electrons is transferred to nitrogenase through ferredoxin or flavoprotein. Nitrogenase uses high-energy electrons and ATP produced by photophosphorylation to reduce proton $H^+$ produced by intracellular metabolism of carbon sources to $H_2$. The hydrogen source provides $H^+$, which is used by nitrogenase [20]. The carbon source is the substance that can provide carbon nutrients for the organism through metabolism during the growth process of photosynthetic bacteria. By comparing the sources of C, H, and N in stems and leaves, the contents of C, H, and N in stems were shown to be 3.39%, 38.01%, and 4.95%, respectively, and in leaves were 4.01%, 41.71%, and 5.25%. The content difference was very small and had little effect on hydrogen production performance. However, the contents of cellulose, hemicellulose, and lignin in stems and leaves varied greatly and were easily degraded into reducing sugars for hydrogen production. The cellulose and hemicellulose content in stems and leaves were 44.6% and 39.1%, respectively. The content of stems was 5.5% higher than that in leaf, and the lignin content of stem was 11.4%, which was 2% lower than that of leaf. Thus, the analysis of the physicochemical properties showed that sweet potato stem was more suitable for PFHP than leaf.

### 3.2. Comparison of Hydrogen Production Capacity of Sweet Potato Stem and Leaf

The hydrogen production capacity was assessed by cumulative hydrogen yield. Hence, the processes of PFHP from sweet potato stem and leaf were monitored.

Biological hydrogen production by photo-fermentation is a complex biochemical transformation process. The pH of hydrogen production feed solution has a significant effect on the metabolic pathway of hydrogen producing microorganisms. Too high or too low pH can cause photosynthetic bacteria to become inactivated and even die. Lignocellulosic biomass

degraded carbohydrate is a small molecule that can be utilized by the microorganisms in the action of cellulase. Therefore, the determination of the reducing sugar concentration of the hydrogen-producing feed liquid in the photo-fermentation biological hydrogen production process can reflect the enzymatic degradation of hydrogen-producing raw materials and the conversion of reducing sugars. Low ORP in the hydrogen production process is a necessary condition for bacterial growth and hydrogen production activities. ORP reflects the net balance of reducing equivalents in the cells of hydrogen-producing bacteria. When the reducing power is higher than the oxidizing power, the ORP will decrease, but on the contrary, the ORP will increase. The reducing power mainly comes from the degradation of the substrate by microorganisms and the growth and reproduction of bacteria. The production of oxidizing power is mainly the dissolved oxygen in the fermentation broth. When the redox status rises, it indicates lack of the main growth carbon source in the growth substrate [21]. Hence, parameters such as hydrogen production rate (HPR), cumulative hydrogen yield (CHY), pH, reducing sugar concentration (RSC), and oxidation–reduction potential (ORP) were determined during the PFHP process. The variations of the process parameters are shown in Figure 2.

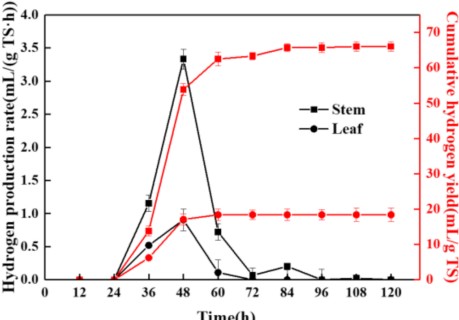

(**a**) Cumulative hydrogen yield and hydrogen production rate

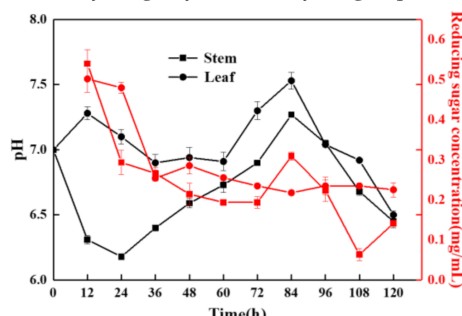

(**b**) pH and reducing sugar concentration

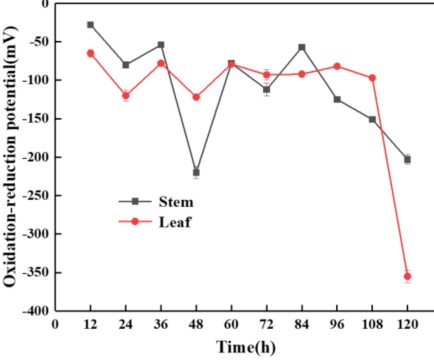

(**c**) Oxidation-reduction potential

**Figure 2.** Variations of cumulative hydrogen yield and hydrogen production rate (**a**), pH and reducing sugar concentration (**b**), and oxidation–reduction potential (**c**) in the PFHP process.

It can be seen from Figure 2a that the hydrogen production rate and cumulative hydrogen yield trends of stem and leaf were basically the same. Within 24–60 h, CHY increased sharply, and the peaks of HPR appeared at 48 h, which were 3.33 mL/(g TS·h) and 0.9 mL/(g TS·h), respectively. The CHY and highest HPR of stem were 66.03 mL/g TS and 3.33 mL/(g TS·h) higher than those leaf. In previous studies, Fan et al. added tea saponin to improve PFHP performance, and the maximum CHY was 58.97 mL/g TS [22]. Thus, stems have potential as PFHP substrates. After 60 h, the PFHP from leaf ceased, while the hydrogen production process stopped at 84 h, which may have been due to the stem producing more reducing sugars, and the reducing sugars were usefully employed in hydrogen production. Related speculations were verified by the determination of pH and reducing sugar concentration.

Figure 2b shows that the pH and reducing sugar concentration changed a lot when stem and leaf were used for PFHP. By observing the changes in PFHP of stem, in the first 60 h, RSC was generally in a downward trend. The reason might be that due to the growth and reproduction of bacteria and the hydrogen production process, reducing sugars were constantly decomposed into small molecular organic acids, resulting in a decline in pH during this period. Moreover, PSB consumed small molecular organic acids in the hydrogen production process, resulting in an increase in pH. Hence, pH increased after 24 h when the hydrogen production phase began. The overall changes of pH ranged from 6.18 to 7.27, which was beneficial for hydrogen producing [8]. Obvious differences of changes in RSC and pH occurred when leaf was used as the substrate. The concentration of reducing sugar in sweet potato leaf was lower than stem, with a peak value of 0.502 mg/mL, which was less than that of stem. Furthermore, due to the increase of pH, the decrease rate of RSC was small, since the environment was not suitable for the survive of PSB. The pH was maintained at a high level, which was nearly neutral. Results indicated that reducing sugar was not effectively utilized by PSB, and hence the decrease of pH was small, and the RSC fluctuated to a certain extent after 24 h. The findings further elucidated that stem was more suitable for PFHP.

Figure 2c shows the ORP changes in the process of PFHP from stem and leaf. It can be seen that the ORP fluctuated slightly in the first 36 h, indicating that bacteria were mainly growing and reproducing at this stage. Within 36 to 60 h, PSB growth and metabolism were vigorous, and the reducing sugar in the fermentation broth was consumed to produce a large amount of reducing power, which resulted in a rapid reduction of ORP. As the fermentation continued, the ORP had an upward trend; it may be that the PSB entered a stable phase from the logarithmic growth phase. The growth and metabolism rate of PSB were slow, and the produced reducing power was used to produce a large amount of hydrogen. In the later stage of fermentation, the overall ORP showed a decrease, because a large number of metabolic inhibitors were produced in the fermentation broth, and the concentration of the available substrate was reduced. Hence, the hydrogen-producing microorganisms gradually decayed, PSB died, and the PFHP process ceased.

### 3.3. Kinetic Analysis of PFHP

The Gompertz model was adopted and conducted to better compare the hydrogen production performance of sweet potato stem and leaf and to determine the lag time of the PFHP process, which can indicate the transition of PFHP from the cell growth to the hydrogen production phase [18]. The kinetic parameters and fitting results are shown and listed in Figure 3 and Table 2.

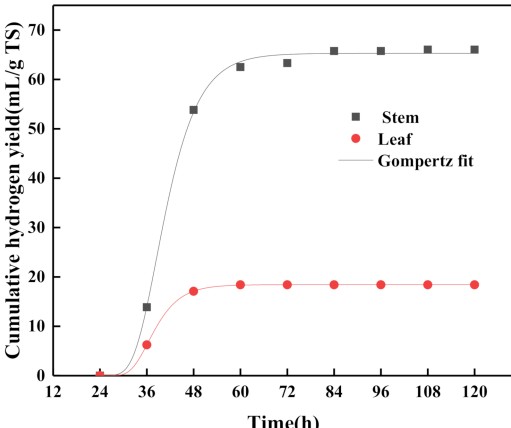

**Figure 3.** Fitting curves of modified Gompertz model adopted for PFHP kinetic analysis.

**Table 2.** Kinetic parameters of modified Gompertz model.

| Group | $P_{max}$ (mL/g TS) | $R_{max}$ (mL/(g TS·h)) | λ | $R^2$ |
|---|---|---|---|---|
| Stem | 65.28 | 4.08 | 32.66 | 0.9988 |
| Leaf | 18.42 | 1.51 | 31.88 | 0.9998 |

A higher correlation coefficient ($R^2 > 0.99$) value was obtained at each group, indicating all data fit well. Comparing the data of stem and leaf fitted by hydrogen production kinetic analysis, the $P_m$ of stem was 3.54 times that of leaf, and the $R_m$ of stem was 2.7 times that of leaf. Thus, the hydrogen production capacity of stem was greater than that of leaf. Lag time of stem and leaf was shown and both began to produce hydrogen at nearly 30 h.

*3.4. Comparison the Energy Conversion Efficiency of Stem and Leaf*

The energy conversion efficiency of the substrate refers to the ratio of the energy converted into hydrogen by the substrate in the hydrogen production experiment to the total energy of the substrate. Energy conversion efficiency of sweet potato stem and leaf for PFHP were calculated using Equation (3). The energy conversion efficiency of stem and leaf was calculated to be 2.01% and 0.56%, respectively. This was similar to the conclusion by Zhang et al., where they reported a maximum energy conversion efficiency of 2.19% by microwave irradiation pretreatment [23]. Stem has a higher energy conversion efficiency than leaf, and an increase of 258.93% in energy conversion efficiency was obtained when stem was analyzed by PFHP. Hence, stem had a higher potential for hydrogen production than leaf.

*3.5. One-Way Analysis of Variance of the Results of Stem and Leaf*

Variations in the cumulative hydrogen production, energy conversion efficiency, $P_m$, and $C_rI$ among stem and leaf were analyzed by one-way analysis for examining the effect on PFHP. The *p*-value refers to significance, whereas the F-value is used to determine significance. When the F-value is larger, it is an indication that the inter-group variance is the main source of variance, and the effect is more significant. On the contrary, if the F-value is smaller, showing that random variance is the main source of variance, the effect is less significant. As shown in Table 3 all *p*-values were greater than 0.05, indicating that the variances were homogeneous and that one-way analysis of variance could be used. The F values of cumulative hydrogen production, energy conversion efficiency, $P_m$, and $C_rI$ of stem and leaf were 2.923, 8.845, 2.964, and 6.133, respectively. The order affected by stem and leaf was energy conversion efficiency $> C_rI > P_m >$ cumulative hydrogen production, as concluded by comparing F-values. This was mainly attributed to the different structure of stem and leaf. In summary, PFHP was significantly different for the different substrates (stem and leaf).

**Table 3.** Summary of one-way analysis of variance.

| Factors | Cumulative Hydrogen Production | | Energy Conversion Efficiency | | $P_m$ | | $C_rI$ | |
|---|---|---|---|---|---|---|---|---|
| | F-Value | *p*-Value | F-Value | *p*-Value | F-Value | *p*-Value | F-Value | *p*-Value |
| Stem and leaf | 2.923 | 0.229 | 8.845 | 0.097 | 2.964 | 0.227 | 6.133 | 0.132 |

## 4. Conclusions

Different parts of plants, such as stems and leaves, often differ in physicochemical properties, which significantly affect the bio-chemical transformation effect. The hydrogen production capacity and energy conversion efficiency of sweet potato stem and leaf were compared. Stem was found to be more suitable for PFHP. Elementary analysis, XRD, and SEM were employed as techniques to determine the hydrogen production potential. The content difference of C, H, and N were very small, which caused little effect on hydrogen production performance; however, the cellulose and hemicellulose content in stems and leaves were 44.6% and 39.1%, respectively. Higher cellulose and hemicellulose content and loose structure indicted higher hydrogen production capacity. The CHY and highest HPR of stem were 66.03 mL/g TS and 3.33 mL/(g TS·h), respectively. The $P_m$ and energy conversion efficiency of stem were 65.28 mL/g TS and 2.01%, respectively, which were 3.54 and 4 times those of leaf. The raw materials source of hydrogen production can be expanded by using vine waste as a substrate. Likewise, vine waste can also be effectively attained for use.

**Author Contributions:** Writing—Original Draft Preparation, H.Z.; Methodology, T.L.; Validation, S.L.; Software, S.Z.; writing—review and editing, Y.L.; investigation, Q.Z.; Funding Acquisition, Z.Z. All authors have read and agreed to the published version of the manuscript.

**Funding:** This research was funded by the National Natural Science Foundation of China (51806061), and the National Key Research and Development Program (2018YFE0206600).

**Institutional Review Board Statement:** This research received no external funding.

**Informed Consent Statement:** Not applicable.

**Acknowledgments:** This study was supported by the National Natural Science Foundation of China (51806061), and the National Key Research and Development Program (2018YFE0206600).

**Conflicts of Interest:** The authors declare no conflict of interest.

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
