# Peer review of "Study on Comparisons of Bio-Hydrogen Yield Potential and Energy Conversion Efficiency between Stem and Leaf of Sweet Potato by Photo-Fermentation"

_fermentation, doi:10.3390/fermentation8040165_

Round 1
Reviewer 1 Report
In this study, the authors analyze hydrogen production from fermentation of sweet potato stem vs. leaf. The results from hydrogen production from stem is compared to that from leaf feedstock. The introduction adequately justifies the research, and in general the conclusions broadly follow from the results. Specific analytical techniques presented are elemental analysis, SEM, XRD and gas chromatography for hydrogen production quantification, in addition to kinetic model fitting. Based on their analyses, the authors conclude that sweet potato stem biomass was more suitable for fermentative hydrogen production. Please see below for a few specific remarks:
Page 1, line 32: Please consider clarifying here that you are referring to the combustion of hydrogen as being carbon-neutral (because significant CO2 emissions can occur from hydrogen production with natural gas, for example).
Page 1, line 40: I understand that PFHP is defined in the abstract, but it may be a good idea to define it again here as this is its first appearance in the main text.
Page 2, line 44: If possible, please (very briefly) give some more information about what is meant by “mild conditions” here.
Page 3, line 111: Was an external calibration necessary to determine the volume fraction of H2 detected by the GC? Apologies if there is something I have overlooked or am misunderstanding.
Page 5, Table 1: Please double check the data entry here, the values of about 4% for carbon seem low.
Page 8-9, Table 3: This Table is appearing to me to be split over two pages. It may be worth trying to fix the formatting here so the table is wholly on one page. Also, I am having some difficultly understanding how these results imply significant differences given the p and F values. Which data are being compared for stem vs. leaf? Can more details be provided here?
Author Response
Thanks for your suggestion. We've made corresponding modifications according to your suggestions and answered your questions one by one.

Reviewer 2 Report
The write-up and explanations present significant inconsistencies. I have provided edits that I hope will make it better. I suggest that the suggestions be incorporated and possibly have a native English speaker proofread the revised manuscript before re-submitting. Good luck

Author Response
Thanks for your suggestion. We have revised the manuscript according to the editing you provided.

Reviewer 3 Report
The manuscript deals with experimental activities and an overview of Study on comparisons of bio-hydrogen yield potential and energy conversion efficiency between stem and leaf of sweet potato by photo-fermentation with the scope of Fermentation journal. The authors should improve the readability and scientific soundness, the manuscript cannot be accepted in this present form, please carefully revise it to improve the quality, the reviewer has been highlighted several open points below:
- The abstract should be rewritten. Answer the questions: What problem did you study and why is it important? What methods did you use? What were your main results? And what conclusions can you draw from your results? Please make your abstract with more specific and quantitative results while it suits broader audiences.your findings Introduction:
This section should be extended, please consider focus also on the exciting technologies that can be used as a bridge between your findings and the recent published articles on photo-fermentation. The frememntation is connected to biogas and biomethane production. Add the proper sentence, consider recent published article: https://doi.org/10.1016/j.jclepro.2021.127404, Moreover add the state on novelty aspect, shoulfd be more visible in the Intro section -
Results and Disscision:
The results section should be improved reporting more detailed analysis, the reviewer suggestions are to perform detailed analysis about the statistic data. Add the proper statistic procedure. Moreover this section should be extended. Add the comparison of your resu;yts with another reserchers in presented subject. - I do not understand section: 3.5. One-way analysis of variance of the results of stem and leaf. Add tyhe prosper disscusion and more detailed results.
- Conclusion in 5 sentences for this experiment is brief. Add the proper conclusions, extend it. Really could your results be summarized in 5 sentences? Those are valuble experimental studies and should be properlly expressed.
Author Response

(The authors gave the same response as above.)

Round 2
Reviewer 3 Report
The manusucript in present form is redy for publication in Energies.
Author Response
Thanks for your suggestion. We have made modifications according to the revision opinions of reviewers
